# Translating Evidence for a Mediterranean-Style Dietary Pattern into Routine Care for Coronary Heart Disease and Type 2 Diabetes: Implementation and Evaluation in a Targeted Public Health Service in Australia [note 1]

**DOI:** 10.3390/healthcare13050506

**Published:** 2025-02-26

**Authors:** Hannah L. Mayr, Lisa Hayes, William Y. S. Wang, Eryn Murray, Jaimon T. Kelly, Michelle Palmer, Ingrid J. Hickman

**Affiliations:** 1Department of Nutrition and Dietetics, Princess Alexandra Hospital, Brisbane, QLD 4102, Australia; eryn.murray@health.qld.gov.au (E.M.); i.hickman@uq.edu.au (I.J.H.); 2Centre for Functioning and Health Research, Metro South Hospital and Health Service, Brisbane, QLD 4102, Australia; 3Greater Brisbane Clinical School, Faculty of Medicine, The University of Queensland, Brisbane, QLD 4072, Australia; lisa.hayes@health.qld.gov.au (L.H.); william.wang@uq.edu.au (W.Y.S.W.); 4Department of Diabetes and Endocrinology, Princess Alexandra Hospital, Brisbane, QLD 4102, Australia; 5Department of Cardiology, Princess Alexandra Hospital, Brisbane, QLD 4102, Australia; 6Centre for Online Health, Faculty of Medicine, The University of Queensland, Brisbane, QLD 4102, Australia; jaimon.kelly@uq.edu.au; 7Centre for Health Services Research, Faculty of Medicine, The University of Queensland, Brisbane, QLD 4102, Australia; 8Department of Nutrition and Dietetics, Logan Hospital, Logan, QLD 4131, Australia; michelle.palmer@health.qld.gov.au; 9The University of Queensland ULTRA Team, Clinical Trial Capability, Centre for Clinical Research, Herston, Brisbane, QLD 4006, Australia

**Keywords:** Mediterranean diet, health services, nutrition, cardiovascular diseases, diabetes mellitus, implementation science, research translation

## Abstract

**Background**: A Mediterranean-style dietary pattern (MDP) is embedded across coronary heart disease (CHD) and type 2 diabetes (T2D) clinical guidelines. However, MDP evidence has not consistently been translated into practice. This study aimed to develop, integrate and evaluate implementation strategies to support clinicians in translating MDP evidence into routine care for CHD and T2D in the local context of a public health service. **Methods**: This study documents the implementation and evaluation phases of a broader knowledge translation project guided by the Knowledge-to-Action cycle. Multi-disciplinary clinicians in the cardiology and diabetes services of two large metropolitan hospitals and a post-acute community service were targeted. Strategies were prioritised utilising theory and stakeholder engagement and included facilitation, building a coalition, the engagement of clinical champions and local opinion leaders, educational meetings, consensus discussions, sharing local knowledge, consumer consultation, and the development and distribution of education materials. Surveys were conducted with clinicians and patients of targeted services to evaluate the reach, acceptability, feasibility, adoption and perceived sustainability of MDP in practice. **Results**: In total, 57 clinicians (7 dietitians, 29 nurses/diabetes educators, 15 doctors and 6 other allied health professionals) and 55 patients completed post-implementation evaluation surveys. The majority of clinicians agreed an MDP is appropriate to recommend in their clinical setting (95%), and most of the time/always their advice (85%) aligns. Education sessions were attended by 65% of clinicians, of which the majority indicated improved knowledge (100%) and change in practice (86%). Factors deemed most important to maintaining an MDP approach in practice were hard-copy education materials (85%) and access to a dietitian (62%). Of the patients who had received care from a dietitian of targeted services (n = 32, 58%), 100% recalled having discussed ≥1 MDP topic and 89% received education material. Of the patients who had received dietary advice from non-dietetic clinicians (n = 33, 60%), 67% recalled having discussed ≥1 MDP topic and 70% received education material. **Conclusions**: Targeted and theory-informed implementation reached the majority of surveyed clinicians and patients, and positively influenced the adoption, acceptability and feasibility of an MDP approach in routine care. Ongoing sustainability strategies are crucial with rotating clinician roles.

## 1. Introduction

Coronary heart disease (CHD) is the leading cause of death and disease burden worldwide [1]. Diabetes doubles the risk of cardiovascular disease, and people with type 2 diabetes (T2D) have a higher risk of CHD mortality [2,3]. Poor diet is a major contributing risk factor to both CHD and diabetes, and approximately 50% of cardiovascular deaths are attributable to this risk factor [4,5]. There is significant overlap in therapeutic approaches to managing these frequently comorbid conditions, with improving dietary habits considered to be a core component of self-management for both CHD and T2D [6,7,8,9].

Dietary practice guidelines now target recommendations for overall diet quality to improve cardiometabolic health more broadly. These dietary pattern recommendations focus on the balance, variety and combination of foods with a multimorbidity lens, and have shifted away from recommendations based on a single nutrient only (e.g., low-fat or low-carbohydrate) [9,10,11]. A Mediterranean-style dietary pattern (MDP) is the most studied dietary pattern worldwide, and there is broad evidence from randomised controlled trials and prospective observational studies that supports its use for both the prevention and the management of a range of chronic conditions, including reduced risk of primary and secondary cardiovascular events, reduction in atherosclerosis progression, improvements in glycaemic control in people with T2D, and reduced incidence of T2D [12,13,14,15,16]. Consequently, following an MDP is recommended as an evidence-based therapy for the prevention and/or management of CHD and T2D across clinical practice guidelines internationally [7,9,10,11,17,18,19]. This dietary pattern is predominantly plant-based, and the key principles include the following: the regular intake of vegetables, fruit, wholegrain breads and cereals, legumes, nuts and seeds, healthy oils (particularly extra virgin olive oil), herbs and spices, fish and seafood, and unflavoured dairy foods (including fermented dairy foods); the moderate intake of lean poultry and eggs; a focus on home cooking or food preparation; and limited intakes of red and processed or fatty meats, and other highly processed foods, sweets and sugary beverages [9,10,13,20].

Despite strong evidence and inclusion in guidelines, the uptake of a foods-focused MDP in routine care, particularly in non-Mediterranean geographical locations, has been poor [21,22]. Furthermore, the dietetic workforce may not be sufficiently confident to translate evidence into practice [23], and other clinicians, such as nurses and doctors, who play a role in delivering nutrition care to people with chronic disease, are often not adequately skilled to do so [24,25,26]. Services for CHD and T2D are also typically delivered in condition-specific siloed clinics, which leads to people with multi-morbidity accessing multiple clinics at risk of and receiving fragmented and contradictory dietary management advice [24], which can make it more challenging for patients to adhere to recommendations [27]. Recent qualitative and survey studies in a non-Mediterranean context have explored dietitians’ views regarding counselling on an MDP in chronic disease management; they demonstrated that adoption is variable, with key influences on practice related to knowledge, awareness of evidence, attitudes, practical skills and resources, and perceived patient interest [21,28,29]. However, no prospective implementation research has focused on determining if and how multidisciplinary clinicians can be supported to consistently recommend this approach in routine clinical practice. There is a need for implementation science methods to facilitate the translation of MDP evidence into the usual care of patients with CHD and T2D in multicultural populations, with the involvement of all clinicians who have a role in dietary care.

Therefore, the current study aimed to develop, integrate and evaluate implementation strategies using implementation science methodology to support clinicians in translating an evidence-based MDP approach into their routine care for people with CHD and/or T2D in an Australian public hospital and community health service context.

### Study Background and Context

The current study forms a component of a broader translational research project, which utilises a pre–post-implementation study design guided by the Knowledge-to-Action Cycle [30]. Here, the ‘knowledge’ is broad evidence that supports the use of an MDP in CHD and T2D prevention and management, and the goal ‘action’ being its integration in routine care across targeted health services. The components of the pre-implementation phase of the project have been reported previously [21,24,31]. In brief, firstly, the problem was identified via the ‘know/do gap’, determined in the Australian setting by conducting a national survey of dietitians treating patients with cardiovascular disease and/or type 2 diabetes (n = 182, conducted in late 2019) [21]. Secondly, to ‘adapt knowledge to local context’ and ‘assess barriers/facilitators to knowledge use’, qualitative interviews were conducted with multidisciplinary clinicians working in CHD or T2D patient care within hospitals and/or associated post-acute chronic disease services of the Australian metropolitan public health service where this project is being undertaken (n = 57, conducted in early 2020) [24,31]. The survey and interview data were mapped to the Theoretical Domains Framework (TDF) [32], with 8 of the available 14 domains considered to have influence on dietary care provided by the relevant clinicians. Key pre-implementation findings and TDF domains are summarised in Table 1.

## 2. Materials and Methods

The current study reports on the next two project phases:Implementation (aligned to Knowledge-to-Action framework stage ‘Select, tailor and implement interventions’) with detailed methods regarding the target routine healthcare setting described in Section 2.1 and implementation strategies in Section 2.2 and Table 2;Evaluation (aligned to Knowledge-to-Action framework stages ‘Monitor knowledge use’ and ‘Evaluate outcomes’ with methods described in Section 2.3, Section 2.4, Section 2.5 and Section 2.6).

The specific objectives of the evaluation were to measure the following: (i) penetration of the implementation strategies to targeted clinicians; (ii) clinicians’ perceived acceptability, feasibility, and sustainability of practicing an MDP approach within routine diet-related care, and (iii) uptake of practice (i.e., adoption) of an MDP approach within routine diet-related care from the perspective of both clinicians and patients. It was hypothesised that implementation strategies would reach the majority of targeted multidisciplinary clinicians and improve self-reported knowledge, skills, access to practice materials, and perceived feasibility and acceptability of the MDP approach, and therefore facilitate the adoption of principles of this dietary pattern in clinicians’ routine care of patients with CHD and T2D.

The study has been reported according to the Standards for Reporting Implementation Studies (StaRI) statement [33].

### 2.1. Target Services and Clinicians

Based on project resources, priorities established from the pre-implementation data and initial executive stakeholder engagement, four services within the broader public health service were targeted for implementation. From Hospital 1 (tertiary hospital with 1074 beds in 2023), this included diabetes (diabetes specialist outpatient clinics) and cardiology (inpatient units, specialist outpatient clinics and nursing led ‘Heart Recovery Service’ delivered across inpatient, outpatient and cardiac rehabilitation program settings). From Hospital 2 (major hospital with 520 beds in 2023), this included cardiology (inpatient units, specialist outpatient clinics and nursing led heart failure service). The fourth service was a community chronic disease service (two clinics including diabetes and cardiac rehabilitation program); this service can receive referrals from the two hospitals as well as other public and private hospitals and clinics. Both inpatient and outpatient/rehabilitation cardiology settings were targeted as pre-implementation interviews demonstrated that clinicians provide healthy eating advice and associated education materials to patients with CHD in both contexts. Across the four services, the implementation was directed to CHD and/or T2D patient care, as well as clinicians from all disciplines who identified any role related to diet.

### 2.2. Implementation Strategies

The planning and execution of implementation strategies took place over 18 months (between May 2021 and October 2022), including research grant funding being sourced for a project facilitator. Outside of the facilitator time, all activity was implemented within current models of care and resource allocations. During initial planning, a core multidisciplinary project team was established that was both strategic and opportunistic (see Table 2 strategy ‘Build a Coalition’). The project team were informed of the pre-implementation findings and brainstormed potential implementation interventions, which considered nuances of the target services. To prioritise the utilisation of implementation strategies that were evidence-based and targeted to the specific identified local context gaps and barriers, the CFIR-ERIC Implementation Strategy Matching Tool [34] was additionally applied (this tool matches Consolidated Framework for Implementation Research barriers to Expert Recommendations for Implementing Change strategies, see https://cfirguide.org/choosing-strategies/ accessed on 25 January 2022). The key identified barriers were selected from the tool’s options, and a prioritised list of strategies to consider was then generated, with the top 15 strategies produced by the tool considered (see Appendix A for strategy list and definitions) [35]. Listed strategies that were deemed to have already been completed in the pre-implementation phase included ‘assess for readiness and identify barriers and facilitators’ and ‘conduct local needs assessment’.

Table 2 provides details on the executed implementation activities and adaptations within and across target services, which are mapped to the relevant ERIC-prioritised strategy and TDF domains being addressed. The executed activities were both iterative and opportunistic dependent on the varied responsiveness and capacity of clinicians across the engaged services, the project team and resource availability. In summary, key implementation activities included the following: having a dedicated project facilitator with MDP clinical and research expertise and knowledge translation training; identifying and utilising clinical champions and local opinion leaders across the target services; developing and/or distributing education materials for staff and patients in hard copy and electronic formats, which covered key evidence and guidelines, dietary principles and practical strategies (including embedding in orientation or caseload material for rotating staff if able); conducting education sessions and consensus discussions with clinical teams within routine education rosters (these were led by the facilitator and involved clinical dietitians); and obtaining consumer feedback on the materials and appropriate ways for clinicians to counsel on the principles of an MDP. Based on feedback from clinicians and consumers, some of the endorsed education materials and practical strategies taught used the terminology of a ‘heart healthy’ dietary pattern, rather than referring to a ‘Mediterranean’ dietary pattern. This terminology was deemed more appropriate for some clinical settings where patients are from diverse cultural backgrounds, and aligns with the National Heart Foundation of Australia recommendations [10]. A description of key challenges faced has been included in Table 2. Project activities were required to adapt to ongoing health service responses and staffing stressors related to the COVID-19 pandemic.

### 2.3. Post-Implementation Evaluation Surveys

To pragmatically evaluate the practical outcomes of the implementation, quantitative surveys were conducted with clinicians and patients of the targeted services. The survey questions were developed specifically for this study through consultation with the research team and were informed by pre-implementation interview findings, implementation strategies and adaptations, and a pre-defined Implementation Outcomes Taxonomy with an associated online repository of tools, which were referred to in order to inform survey question design [36]. The implementation outcomes of interest to be measured by the survey questions were as follows:Penetration (reach and impact of the implementation to targeted clinicians and patients);Acceptability (clinician satisfaction with aspects of the intended MDP approach in practice, including credibility, content, and complexity);Feasibility (clinician perception of the actual fit or utility of the practice within the setting);Adoption (clinicians’ action to employ the MDP approach in practice, with confirmation from patients);Sustainability (perceived ongoing strategies required for routinisation by clinicians).

For both surveys, question logics were used, with the number of questions asked dependent on survey responses to questions related to exposure to implementation strategies, and either the clinician’s role or patient’s medical history and service access. The study was approved by the Metro South Human Research Ethics Committee (approval HREC/2022/QMS/87976) and all participants provided informed consent. Participants were recruited over a 6-week period staggered across target services between November and December 2022. For both clinicians and patients, the study invitations and information did not specifically refer to an MDP approach so as to avoid biasing those who chose to participate. Patients were offered to enter a prize draw to win one of five supermarket vouchers (value AUD 100) if they participated in the survey.

### 2.4. Clinician Survey Recruitment and Questions

The clinician survey was anonymous and administered electronically via Qualtrics^XM^ (Qualtrics, Provo, UT, USA). The eligible clinicians were as follows: (1) a dietitian, doctor (medical officer, advanced trainee or resident), clinical nurse, diabetes educator or other allied health professional; (2) employed by the health service working within at least one of the four targeted services during the implementation and/or evaluation period; and (3) involved in providing diet-related care or referral to dietetics services for patients with CHD and/or T2D. A total of 253 clinicians (7 dietitians, 157 nurses or diabetes educators, 71 doctors and 18 other allied health professionals) were invited to participate via email either by a project team member or clinical champion (Figure 1). The emails were sent to broader cardiology and endocrinology staff groups, hence not all these invited clinicians would have fulfilled the eligibility criteria related to providing diet-related care for relevant patients. Where feasible, recruitment reminders were included in routine clinical team meetings or via flyers in common areas. The target participation number was 50, based on engagement in the pre-implementation study interviews and a recruitment rate of approximately 50% of the estimated number of clinicians that had engaged with the implementation strategies in routine care.

Firstly, the survey questions captured the following: clinical role, healthcare setting and demographics; perceived role related to diet; recall of exposure to or involvement in implementation activities and learnings/feedback. Secondly, the questions asked whether the current dietary advice or materials used align with the implemented dietary pattern approach, and the perceived acceptability and feasibility of recommending this approach in practice for relevant patients; prior to these questions, a list of core dietary pattern principles was provided (see Appendix A). Questions relating to specific information materials provided were presented as options in a checklist. The survey was piloted by a clinician to inform readability, flow and appropriate length. The survey was expected to take approximately 15 min for dietitians and 10 min for other clinicians.

### 2.5. Patient Survey Recruitment and Questions

The patient survey could be completed in an electronic format in Qualtrics, or via hard copy or telephone (and entered into Qualtrics manually by the project lead). Eligible patients met the following criteria: (1) adults (aged ≥18 years) diagnosed with at least one of CHD or T2D, who (2) had attended at least one outpatient consultation or group program session with a doctor, nurse, diabetes educator and/or allied health professional within targeted cardiology or diabetes services within the month prior to or during the evaluation period. The following exclusion criteria were applied: unable to read English language, presence of any palliative care order, pregnancy or breastfeeding, or presence of any of the following comorbid conditions that significantly impacts on dietary prescription—chronic organ failure, type 1 diabetes, prior bariatric surgery, inflammatory bowel disease, recent solid organ transplant, receiving dialysis treatment, or recent diagnosis of malnutrition.

Potentially eligible patients were identified through previous or upcoming clinic lists of the targeted services. Patients were screened for having a CHD and/or T2D diagnosis and speaking English. The screening of clinics aimed to capture representation from across targeted services and dietitian accessibility. Relevant patients were then invited to participate by the project lead who was not involved in delivering clinical care, either (i) remotely via text message, with a phone call follow up if the patient had not opted out or completed the survey within 24 h; or (ii) in-person at the time of their clinic appointment. The target participation number was 50, so as to derive similar representation as clinicians (Figure 1).

The survey questions captured the following: demographic characteristics; relevant medical history; target service/s accessed; recall of having received any care related to diet from dietitians or other health professional/s, the nature of the diet-related care received (options for topics discussed or materials provided were presented as a checklist), including whether aspects of the implemented MDP approach were included. The survey was expected to take between 5 and 15 min.

### 2.6. Data Analysis

Survey data were exported from Qualtrics to Microsoft Excel (2016, Microsoft Corp, Redmond, WA, USA), cleaned and coded, then imported to SPSS (2022, IBM Corp., SPSS Statistics for Windows, Version 28.0, Armonk, NY, USA) for descriptive and statistical analysis. Data are reported as n (%) or median and range in the total cohort or by professions. Characteristic data on the invited patients who chose to participate versus not were compared using Chi-squared or independent T-tests. Clinicians’ responses to the key adoption question of how often their dietary advice aligns with the MDP principles were compared between clinician characteristic data categories (age categories; gender; whether born in Australia; duration in health professional role ≤ or >10 years; duration working with target patient group/s ≤ or >10 years) using Chi-squared tests. Chi-squared tests were also used to compare responses to questions related to penetration, acceptability, feasibility, adoption and sustainability between profession groups (dietitians; nurses/diabetes educators, doctors and other allied health professionals). Statistical significance was set at *p* < 0.05.

## 3. Results

### 3.1. Clinician Surveys

A total of 69 clinicians responded to the survey (Figure 1). Here, 12 were excluded; 4 were ineligible (n = 2 were not working in target services and n = 2 did not identify any relevant diet-related role) and 8 responded only to demographic and/or role characteristics questions. Survey response data from 57 clinicians were therefore included (Table 3) with a total response rate of 23% (57 of 253 clinicians emailed who were potentially eligible). The median age was 38 years (range 21 to 62). There was diverse representation from across relevant professions, with seven dietitians, 29 nurses or diabetes educators, 15 doctors and six other allied health professionals. Diet-related roles and usual practices for patients with CHD and T2D are reported in Appendix A.

#### 3.1.1. Penetration

Table 4 reports data related to the reach and impact of select implementation strategies in the total clinician survey cohort and by professional roles. The majority (65%) of clinicians had attended or watched a recording of an education session, of which 100% of attendees reported having learnt something new and 86% felt the session/s resulted in a change to their practice. The majority (67%) of clinicians recalled having accessed one or more of the provided electronic resources. The most accessed were copies of patient education materials (51%) and the education session slide set (42%), and less common were specific publication/s of a study (16%) or guidelines/evidence summaries (27%). Table 4 also reports on the reach of orientation, handover or caseload material related to diet for clinicians who had commenced their role within the prior year (n = 22). Of these, the majority (55%) did not recall receiving any such material related to diet. Just over one-third (36%) indicated they had received such, relating to a Mediterranean-style or heart healthy dietary pattern; 67% of dietitians and 25% of other professions. There was no statistically significant difference between profession groups for these penetration outcome variables (Table 4).

#### 3.1.2. Acceptability

Table 5 reports data related to the perceived acceptability, feasibility and adoption of the Mediterranean-style or heart-healthy dietary pattern approach in practice. There were no statistically significant differences between profession groups for these reported outcomes; however, some notable observed differences are referred to in the text results. All clinicians agreed the approach is credible (i.e., would align with scientific evidence) and 95% agreed this dietary approach is appropriate to recommend in their clinical setting. Just over half (56%) felt the approach is rarely or never too complex to raise or recommend to relevant patients; dietitians were less likely to report the approach as too complex, with 86% responding rarely or never to this question.

#### 3.1.3. Feasibility

Having enough time to raise or recommend this approach in practice differed between clinical settings and by profession. For clinicians working in outpatient/community settings or in cardiac rehabilitation, 59% responded with most of the time or always for this question, compared to 44% of those in an inpatient setting. Doctors had the lowest proportion of participants reporting that they have enough time most of the time/always (10% in an inpatient setting and 47% in an outpatient setting). The majority (59%) of clinicians reported that patients would be able to improve their eating habits to better align with this dietary pattern most of the time of/always. The response rate was similar across professions for this question, except doctors, where 40% responded most of the time/always, 47% sometimes and 13% rarely or never.

#### 3.1.4. Adoption

Eighty percent or more of the clinicians indicated that their advice and diet-related education materials or tools provided would now align to Mediterranean-style or heart-healthy dietary pattern principles. There was a significant association between gender and how often clinicians’ advice aligned to the MDP (95% of females indicated most of the time/always versus 57% of males, *p* < 0.010), but no significant association with any other characteristics variable (Appendix A). The majority indicated that, most of the time/always, they focus on foods or meals more than nutrients or calories (70%) and on what to include more than what to restrict or cut out (57%). The uses of recommended patient education materials are reported in Table 5 (by type and key clinic handouts) and Appendix A (all specific materials including across services and professions). All the dietitians and 79% of other professionals surveyed reported having used one or more. One targeted service, Hospital 1 diabetes outpatient clinics, reported a higher use (100% had used one or more) than in others (73 to 78%). With regard to materials displayed in clinic rooms, the two-page health service Mediterranean-style diet factsheet had been used by more than half the clinicians, including 60% of doctors and 85% of hospital 1 diabetes outpatient clinicians. This handout included a checklist for assessing alignment to 10 key Mediterranean-style eating principles to assist goal setting.

#### 3.1.5. Sustainability (Perceived Requirements)

Clinicians who indicated that their practice aligned with an MDP approach, either in advice or materials provided (n = 53), were asked to identify what they see as being most important to this being maintained (Table 5). Each option was selected by more than half the participants, including access to patient education materials in hard copy (85%) and electronic (60%) formats, access to or relationship with a dietitian (62%), refreshers or updates on practical tools/tips (58%) and refreshers or updates on evidence (52%). For this latter outcome, there was a statistically significant difference between professions, with >80% of dietitians or other allied health professionals identifying refreshers/updates on evidence to be important, versus 58% of nurses and 20% of doctors.

### 3.2. Patient Surveys

The total number of potentially eligible patients invited to participate was 131 (Figure 1). Of these, 14 (10.7%) were initially invited in-person and 117 (89.3%) were initially invited via text message (of which only 12 participants sent an opt-out text reply). A total of 59 patients responded to the survey (59/131, response rate of 45%). Four were excluded as they were ineligible (n = 1 self-reported they did not have a relevant condition despite their medical history indicating CHD diagnosis, n = 1 reported having prior surgery for obesity, n = 1 had type 1 diabetes and n = 1 was receiving palliative care). Survey responses from 55 patients were therefore included (Table 6). The mean age was 61 ± 11 years (range 21 to 87). There was representation of patients from across targeted services and clinical settings. More than half reported having received care from a dietitian/s (58%) and diet-related care from other health professional/s (60%) within a targeted service. There were no significant differences between patients who chose to participate in the survey and the patients who were invited but did not participate with regard to age, history of CHD, history of T2D, and whether they had recently accessed a dietitian in a relevant service (*p* > 0.05).

Table 7 reports on the patient respondents’ recall of diet-related care received. Relevant participants recalled that at least one or more of the topics related to principles of a Mediterranean-style or heart-healthy dietary pattern (which were listed in the survey as multiple-choice options) had been covered by dietitian/s 100% of the time and by other health professional/s 67% of the time. With regard to topics of information materials that aligned with a Mediterranean-style or heart-healthy dietary pattern approach, 89% recalled having received at least one from a dietitian and 70% from another health professional. Patients reported having read some or all of the material provided by dietitian/s 84% of the time and by another health professional 88% of the time. In those who had received diet-related care, 84% reported they learnt something new and 96% reported they had already made dietary change/s.

## 4. Discussion

To our knowledge, the current project represents the first study to use implementation science methods to translate scientific evidence for a Mediterranean-style dietary pattern into routine care for people with CHD or T2D. Targeted implementation was informed by theory, and addressed identified barriers and enablers for clinicians in the local context of hospital and community clinics of a public health service in Australia. Evaluation surveys of clinicians from across dietetics, medical, nursing and other allied health disciplines, as well as patients of targeted cardiology and diabetes services, demonstrated that implementation strategies positively influenced implementation outcomes.

As hypothesised, the targeted implementation strategies improved the clinicians’ perceived acceptability and feasibility of an MDP in the local context of CHD and T2D patient care. The survey findings related to clinicians’ perceptions of the credibility, appropriateness, and complexity of the MDP approach contrast with findings from the pre-implementation consultation phase. At pre-implementation, there was variable understanding and acceptance of Mediterranean diet evidence, where some perceived a lack of supporting clinical trial data, that this approach was not part of current guidelines, or that it would not achieve desired clinical outcomes in patients who lack the ability to follow this dietary pattern [31]. Some clinicians, including dietitians, had expressed concerns with recommending a specific ‘diet’ or using the terminology ‘Mediterranean’ in a multicultural setting [21,31]. Similarly, UK-registered dietitians expressed that, whilst they advocate the principles of the MDP, they do not necessarily call it ‘the Mediterranean diet’ [28], and health professionals in Ireland demonstrated that the sociocultural context and tailoring of communication are important to address regarding the use of an MDP in practice [37]. These barriers were addressed in the educational content and materials delivered, including tips from early adopters and iterative consumer consultation on the appropriateness of materials and dietary advice fed back to clinicians; these adaptations likely supported the perceived acceptability and feasibility. Of note, none of the surveyed clinicians were of Mediterranean background, and hence this study demonstrates the implementation of the approach for clinicians who do not identify with this eating pattern culturally.

Post-implementation surveys highlighted an adoption of the intended practice approach across disciplines with regard to advice or diet-related education materials aligning with a Mediterranean-style or heart healthy dietary pattern, and the data suggest this was impacted by the implementation strategies utilised. The education sessions with associated education materials reached the majority of the surveyed clinicians and reportedly impacted both knowledge (of dietary principles and evidence/guidelines) and actual practice. Notably, all the involved dietitians reported a change in practice, and most reported improved knowledge. Therefore, it cannot be assumed that dietitians have the necessary knowledge and skills to practice this dietary approach, and indeed previous data have demonstrated that clinical dietitians may lack confidence in evidence translation [23]. The education was led by the facilitator, a post-doctoral research dietitian with expertise in the specific topic area and who had experience working in the target setting, which likely enhanced the outcomes. Our pre-implementation clinician interview data suggest that senior or expert direction is required to achieve Mediterranean diet practice adoption [31,38], and prior literature suggests that dietitians look to colleagues and experts in the field as part of evidence-based practice [39]. Furthermore, the facilitator was employed (using research funds) to manage and oversee all project activities. As described in the ERIC compilation of implementation strategies, facilitation involved interactive problem-solving and relied on supportive inter-personal relationships [35].

Achieving engagement with the education sessions and sharing of materials with relevant clinical staff also relied on local opinion leaders and clinical champions (from across professional backgrounds), whose involvement and endorsement was acknowledged. Clinical champions, overlapping with local opinion leaders, represent a highly utilised implementation strategy to improve evidence translation efforts in routine care settings, and it has been suggested that the mechanism operates through two key causal pathways—(i) intention development and (ii) behavioural enactment [40]. Furthermore, delivering tailored sessions within existing routine meeting schedules such as department ‘journal club’ or in-services appeared to be crucial to achieving reach and securing the interest of doctors and nurses without additional burden. In the implementation of an evidence-based program for the self-management of T2D and hypertension, it was similarly demonstrated that program adaptations that conserve staff time and resources and recognise their contribution can increase program effectiveness without jeopardising fidelity [41].

Insufficient time and workload pressure, particularly in the context of competing acute and chronic priorities for complex patients, were identified as barriers to recommending an MDP in a public health care setting through our national and local pre-implementation data collection [21,31,42]. These findings aligned with other existing evidence collections of barriers to dietary care in chronic disease management [26] and the implementation of evidence-based practice in general [43]. The current post-implementation surveys demonstrate that insufficient time continued to be an aspect that challenges the feasibility of incorporating an MDP in usual care for non-dietitians, particularly in inpatient settings and for doctors. The adoption data collected were intentionally focused on whether the intended evidence-based approach was being utilised when dietary advice or materials were provided, and not on how often. Nonetheless, the survey data regarding high use of endorsed patient education materials (reflected in both clinician and patient data) suggests that access to these aided in the adoption of the approach within time pressures. From the range of provided materials, the clinicians, including doctors, were more likely to have accessed patient materials than evidence summaries or study publications. This suggests that in the setting of a busy workforce, once evidence is accepted and trusted, many clinicians are seeking the most practical ways to implement it, rather than supporting academic literature.

Implementing an ‘evidence-based’ dietary approach in chronic disease management is complicated by the fact that there are multiple food and meal preparation components to a healthy dietary pattern, and, as emphasised in current guidelines and by clinical dietitians, individualisation is important [9,10,28]. This makes the intended health professional behaviour change challenging to both support and to measure [44]. Whilst there is no ‘one size fits all’ diet approach for CHD and T2D, our implementation responded to local data showing that many clinicians were providing out of date and inconsistent nutrient- or exclusion-focused recommendations, and that departmental-cultural shifts towards a unified dietary pattern approach that focuses on the inclusion of healthy foods would assist in establishing consistency and aligning with evidence [24,31,38]. Other studies on nutrition evidence translation in health care settings have tended to focus on acute care where, for example, post-operative early oral feeding is implemented—this allows for pre- and post-clinical audit data and a more specific innovation of practice, such as a feeding protocol [45,46]. The current mode of measurement of adoption was pragmatic, including patient recall of having been advised on any of the food-focused principles of an MDP or provided endorsed materials, as the inclusion of any of these was considered to reflect the implementation of the intended approach. Ideally, objective measures of practice could be used [47]. Whilst it was considered, it was not deemed suitable to conduct audits of consultation documentation in medical records, as these may not provide an accurate reflection of the nature of nutrition’s inclusion in patient care. Conducting observation was also not suitable or practical due to potential bias with observed practices and social distancing requirements at the time.

The fidelity of the intended implementation activities (Table 2) and clinician survey data provides some useful insights into impacts on the sustainability of the dietary approach in practice without the project facilitator. Staff turnover is a commonly reported barrier to sustaining healthcare practices [48], and appears to be crucial to address in the current setting. Whilst it was intended for relevant workload orientation/handover material to be updated with summary evidence and practical tools, this was not successfully implemented across the areas and disciplines. Most non-dietitian survey respondents who had recently commenced their workload did not recall having received any orientation related to accepted diet therapies. In some instances, this was related to there being no central documentation or electronic repositories to store the information. The rotation of dietitians was common, and in some services the facilitator was required to repeat the upskilling and sharing of materials despite having intended for this to be encompassed in clinician handover. The clinicians indicated that access to patient education materials, particularly hard copy, would be important. Our needs assessment data [21,24,31,38,42] and other literature [49] suggest that accessible, pragmatic and endorsed patient handouts assist clinicians’ memory and decision processes, guide patient education or goal-setting, and support the prioritisation of lifestyle components within time pressures. Nearly all the surveyed patients who recalled having been provided nutrition materials reported having read at least some of the information. Relationship with a dietitian and refreshers/updates on practical tools were also regarded as important to sustaining the approach; this project supports the interpretation that dietitians are upskilled in MDP and it should be embedded within clinical teams, including providing inter-disciplinary training and direction towards evidence-based materials on a more frequent basis.

Finally, an interesting finding was that the rate of MDP practice adoption was higher in females compared to males. All the surveyed dietitians were female; however, since dietitians only represented a small proportion of the total cohort, this can only partly explain this finding. The implementation research literature shows that gender can impact decision-making, stakeholder engagement, communication strategies and preferences for the uptake of interventions [50], and hence potentially the chosen strategies were unintentionally biased towards female preferences, meaning that future implementation work could consider barriers and enablers specifically in the context of gender.

### 4.1. Strengths and Limitations

Our study is strengthened by its utilization of multiple evidence-based theoretical frameworks, tools or repositories to guide the implementation and evaluation, which was pragmatic by necessity. It is also strengthened by the inclusion of all health professionals who have a role in dietary care. The evaluation is, however, limited by not having comparable pre- and post-survey data; hence, the degree of change or impact on outcomes between pre- and post-implementation is difficult to quantify. Nonetheless, interviews were deemed necessary at pre-implementation to derive a deeper understanding of barriers and facilitators, and were crucial to the development of the targeted strategies. The recruitment strategies used, including support from clinical champions and departmental leaders, ensured that all eligible clinicians were invited. However, the post-implementation data are limited to the respondents who chose to complete the survey in the setting of a busy clinical environment. Whilst 100% of relevant dietitians responded, the total clinician response rate was 23%. It is not known what proportion of the invited clinicians met the full eligibility criteria, and in particular, many of those invited would not have had a diet-related role. Our advertising for survey involvement did not refer to an MDP specifically; however, there is potential that clinicians who engaged in the implementation activities, who made changes to practice or who had already adopted the intended approach were more likely to respond. Patient respondents may also represent a cohort who were more engaged or motivated in relation to diet. However, we did determine that there were no differences in terms of age or proportion of individuals who had seen a dietitian between those invited patients who chose to participate and those who did not. A further limitation is that patients unable to read or speak English were not included. Finally, the surveys were designed specifically by the research team for this evaluation, and whilst they were informed by the Implementation Outcomes Taxonomy and associated online repository of published tools, they did not undergo specific validity testing.

### 4.2. Novelty and Future Directions

This paper reports the first study to use an implementation science methodology to translate evidence related to an MDP into a local healthcare setting. The prospective documentation of implementation strategies and their fidelity, as well as the inclusion of patient surveys to confirm clinicians’ self-reported MDP practice adoption, are novel. Furthermore, a recent scoping review found that very few studies that report on implementation outcomes have done this in relation to implementation strategies [47]; therefore, this study makes important contributions to the body of implementation science literature by reporting on outcomes related to adoption, penetration, acceptability and feasibility as a function of specific implementation strategies used. Aligned with the final phase of the Knowledge-to-Action framework, the sustaining of MDP adoption in practice in this targeted cohort is planned to be evaluated locally. The methods utilised in this study could also be replicated and adapted to other routine healthcare settings, and a larger study of different health systems would be warranted, especially in similar multiethnic settings.

The current project only targeted functions within existing resourcing and clinical models. Our pre-implementation data suggest that broader issues related to optimal dietary care for CHD and T2D exist within this type of setting. These relate to limited dietitian access and follow up, referral processes, and nutrition-related treatment prioritisation [24]. Furthermore, suggestions were made by clinicians during consensus discussions related to expanding opportunities for practical learning with patients (e.g., cooking demonstrations) and reviewing the hospital food service; however, these were not able to be implemented. Key components of intervention studies conducted outside the Mediterranean that achieved high compliance to a Mediterranean-style diet have been summarised to include the following features: dietitian-led, dietary education, goal setting, recipes (simple and affordable), meal plans, food checklists, regular health professional contact, food hampers and cooking classes [51]. Ongoing implementation research, including engagement with diverse consumers of target settings, and policy work is needed to support the innovation and expansion of dietary care models in public tertiary/post-acute specialist services so as to better align with and adapt to these evidence-based strategies, which have been proven to support patient dietary adherence.

## 5. Conclusions

The current implementation science project achieved the improved translation of a Mediterranean-style or heart-healthy dietary pattern into routine care for people with CHD and/or T2D in a public hospital and community health service context in Australia. Across dietetics, nursing, medical, and other allied health disciplines, targeted implementation strategies impacted knowledge, skills and culture, which had a positive influence on the perceived acceptability and feasibility, as well as the adoption, of an evidence-based dietary pattern approach. The Knowledge-to-Action cycle provided an overarching guide to the implementation project phases. However, the overlapping of this cycle with more in-depth theoretical frameworks and tools was necessary to prioritise barriers and enablers in the local context, and to select, tailor, and facilitate appropriate implementation strategies. In the setting of a time-pressured workforce with routine staff turnover, ongoing support for sustainability that does not rely on a dedicated change facilitator is crucial. The engagement of adequately skilled dietitians in clinical teams and access to patient education materials are key strategies to prioritise for ongoing sustainment. Innovations to dietary models of care in tertiary settings that enhance behaviour changes, as well as equity and ease of access, should be further explored.

## Figures and Tables

**Figure 1 healthcare-13-00506-f001:**
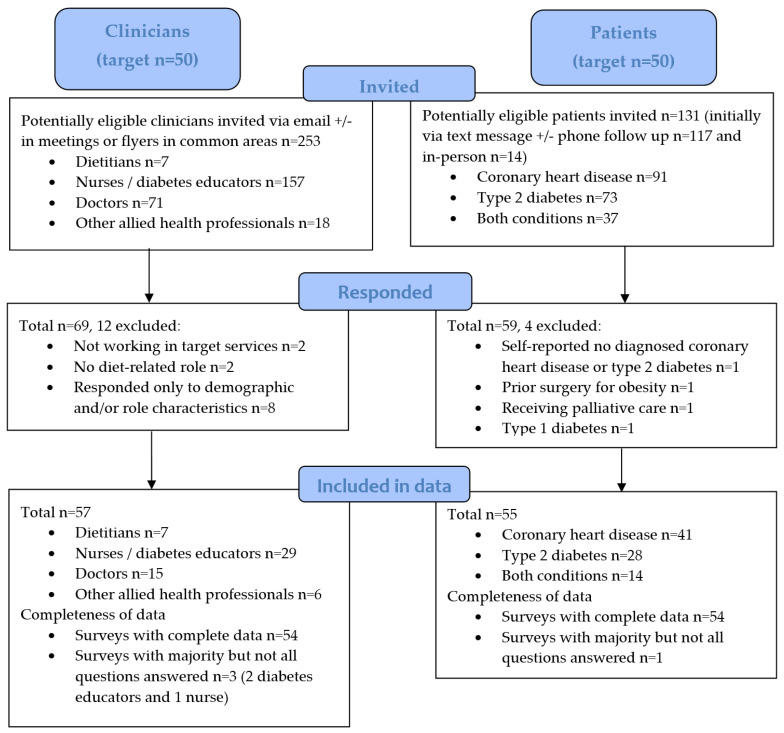
Participant flow diagram of recruitment and data inclusion for clinician and patient evaluation surveys.

**Table 1 healthcare-13-00506-t001:** Summary of local evidence–practice gaps and barriers and enablers to clinicians implementing a Mediterranean-style dietary pattern in coronary heart disease and type 2 diabetes care.

Identified Practice Gaps, Barriers and Enablers *	Relevant TDF Domains
Clinicians from across disciplines are involved in dietary care and referrals Inconsistencies in dietary recommendations to patients between clinicians and conditions, despite patients having multimorbidity Many focused recommendations on single nutrients and what to restrict or exclude from the diet Core food-based Mediterranean-style dietary pattern principles were not part of routine care, particularly for non-dietitians Many clinicians had a poor understanding of or questioned the nutrition evidence and core dietary pattern principles Many perceived the approach to be complicated and potentially not acceptable or feasible to recommend in their setting Most dietitians acknowledged sufficient evidence, but are seeking practice-focused professional development, particularly dietitians in public health services and hospital/secondary care settings Lack of nutrition education and training of non-dietitians with limited knowledge and skills sharing from expert dietitians Many clinicians were time-poor; strategies that target the professional roles and capacity and the busy hospital/secondary care setting, and which are simple to implement, are needed The service culture is important (but was lacking with regard to nutrition), and greater direction from experts and opinion leaders, as well as local consensus, would be highly valued Lack of belief in patient capabilities or interest to improve dietary adherence prevents some clinicians promoting a Mediterranean-style dietary pattern Access to practical and simple patient education materials that are endorsed by reputable bodies or experts are needed	Professional role and identity Knowledge Skills Beliefs about capabilities Environmental context and resources Social influences Beliefs about consequences Memory, attention and decision processes

* Data obtained from national dietitian survey [21] and local interviews [24,31] with dietitians, doctors, nurses, diabetes educators and other allied health professionals of diabetes and cardiology services in a metropolitan public health service. TDF, Theoretical Domains Framework.

**Table 2 healthcare-13-00506-t002:** Implementation activities in targeted diabetes and cardiology services mapped to implementation strategies and theoretical domains.

ERIC Strategy	TDF Domain/s	Description of Activities Executed Within or Across Services Between May 2021 and October 2022	Services Engaged with Activity				Key Challenges During Implementation Period
			1	2	3	4	
Facilitation	Environmental context and resources	Facilitator employed in research assistant role for project management and planning, execution and oversight of all activities - Clinical and research expert in the evidenced-based approach being implemented - Completed knowledge translation training and prior experience with research translation projects in local health service - Supervised and mentored by Senior dietetics Research Fellow	√	√	√	√	- Facilitator office based at H1 with no specific workspace at other sites - Facilitator employed by H1 Dietetics and well known to staff but not embedded or familiar at other sites
Build a coalition	Professional role and identity Environmental context and resources	Project team from across services recruited by facilitator - Interest identified and informed of pre-implementation findings - Agreed to involvement in meetings and support of strategies in relevant services - H1 Senior dietetics Research Fellow with knowledge translation expertise - H1 Dietitian Team Leader managing relevant dietetics workloads - H1 Senior Medical Officer and Director of Diabetes and Endocrinology - H1 Senior Medical Officer and researcher in Cardiology - H2 Dietitian Team Leader and clinical researcher (also connected to CDS dietitian) - University dietetics Research Fellow with chronic disease and telehealth expertise	√	√	√	×	- Dietitian Team Leaders in project team not well linked with cardiology services - H1 Dietitian Team Leader role changed multiple times
Inform local opinion leaders	Social influences Environmental context and resources	Opinion leader/s from relevant services engaged (including some project team members and other medical and nursing service directors)					- Contact with cardiology medical directors typically required via administration staff and not direct - H1 Heart recovery service nurse lead/s changed or was unfilled role
		- Informed of, approved and supported activities with staff and consumer engagement	√	√	√	√	
		- Relevant dietetics Directors and Team Leaders were engaged, including approving involvement of clinical dietitians	√	√	√	√	
Identify and prepare champions	Professional role and identity Knowledge Skills Beliefs about consequences	Clinical dietitian/s consulted on pre-implementation findings and implementation ideas					- Hospital cardiology dietitian roles new graduates and rotated 6–12 monthly; were not heavily embedded nor physically located with inter-disciplinary team - H2 cardiology dietitian funded mostly for heart failure service (limited ward and no outpatient role hence had limited interaction with medical and ward nursing staff) - CDS dietitian changed mid-year
		- Provided with evidence-based literature and practical materials and upskilled on MDP approach as appropriate	√	√	√	√	
		- Contributed to direction and execution of strategies for respective sites with varying degrees of involvement	√	√	√	√	
		- Dietitian handover of project and prior service activities when dietitian rotated or on leave (if not, facilitator repeated as necessary)	√	√	×	×	
		Interested diabetes educator and nurses helped engage clinicians and consumers, provided feedback, and supported access to materials in clinical areas	√	√	√	√	
Develop educational materials	Knowledge Skills Memory attention and decision processes	MDP materials appropriate for use across disease specialities were sourced, created or adapted at varying stages based on consumer/clinician feedback Clinician materials: - Key evidence summaries/position statements from reputable bodies and publications and key result summaries from landmark trials or systematic reviews - Summarised evidence–practice gap findings from pre-implementation interviews - Practical strategies for patient advice, responding to questions or dietitian referrals - Patient education materials - Created 2-page ‘Mediterranean-style diet’ health service factsheet - Facilitator engaged with an established state government dietitian-led initiative *, which supported creation of and/or updates to factsheets and cardiac rehab group presentation - Sourced aligned web-based and printable factsheets from Australian reputable bodies (National Heart Foundation and Baker Heart and Diabetes Institute) - Sourced recommended cookbooks or websites with appropriate recipes - Sourced relevant healthy eating infographics suitable for use as posters - Created slide content on MDP key tips for use in TV waiting rooms of clinics - Collated healthy eating/food preparation videos onto a freely accessible webpage	√	√	√	√	- Patient materials developed needed to be feasible for use within existing models of care and resourcing and hence were largely focused on facilitating knowledge transfer rather than additional behaviour change strategies - Within current resources/models it was difficult to establish a platform for hosting videos and having these accessible to patients
Distribute educational materials	Knowledge Skills Memory attention and decision processes	- The above materials were initially presented to clinicians in education sessions via slide set and hard copies of key factsheets	√	√	√	√	- Some teams did not use a shared electronic file location - Some services lacked a clinical champion willing to restock handouts and facilitator fulfilled, or it was unknown if monitored - TV slide set was created for H1 cardiology outpatient clinic but was never displayed due to staffing issues - In response to clinician feedback a webpage of videos was created and shared October 2022 and relied on clinical champions to share
		- Materials shared electronically for ongoing access with relevant clinical teams	√	√	√	√	
		- Hard copies of preferred short factsheet/s displayed in clinic rooms	√	√	√	√	
		- TV slides displayed in clinic waiting rooms	√	×	×	×	
		- Healthy eating posters or copies of materials displayed in waiting areas or gym	×	√	√	√	
		- MDP included in orientation or caseload/handover material for:					
		○ dietitians	√	√	×	×	
		○ doctors	√	×	×	NA	
		○ nurses/diabetes educators +/− other allied health	×	√	×	×	
Conduct educational meetings	Knowledge Skills Beliefs about capabilities Beliefs about consequences	Education sessions delivered to clinical teams by the facilitator - In-person, online or hybrid format within routine education rosters (multiple conducted for cardiology ward nurses to capture part time/shift staff) - Some recorded with ongoing access - Tailored to clinical team and time allotted - Presented what has been described in ‘clinician materials’ and showcasing most relevant patient education materials - Integrated time for questions and initial feedback or concerns	√	√	√	√	- Attendance and roles of attendees not feasible to capture, hence true exposure unknown - Using an MDP approach is not a single simple behaviour to teach and its integration within patient-centred care can be complex - * Joined only for 1 nurse education session
		Clinical dietitian delivered part of practical section if available/confident	√	P *	×	√	
Conduct local consensus discussions	Environmental context and resources Social influences Beliefs about capabilities	Discussions were facilitated with teams ~4–6 months after initial education session - Reminded of delivered education content and materials - Sought feedback on use of approach and ideas of ongoing adaptation of strategies	√	×	P	√	- No follow up sessions were scheduled with H1 cardiology or the H2 cardiology ward nurses - * Joined for heart failure service discussion, not medical team
		Clinical dietitians were involved as feasible	√	NA	P *	√	
Create a learning collaborative	TDF Domains and activities for this strategy are deemed to overlap with those utilised for ‘distribute education materials’, ‘conduct educational meetings’ and ‘conduct local consensus discussions’						
Obtain and use consumers and family feedback	Beliefs about consequences	Consumers from target services were consulted - Feedback on preferred timing for, format and nature of dietary information and appropriateness/interest in ‘Mediterranean-style’ dietary pattern - All newly developed and/or endorsed patient education materials involved a formal consumer review process	√	√	×	√	- Difficult for facilitator to consult consumers in-person with social distancing requirements, which delayed execution of these strategies
		In consensus discussions, clinicians were facilitated to provide feedback that consumers had given on approach and materials, which informed adaptions and ongoing education	√	×	√	√	
Capture and share local knowledge Identify early adopters	Social influences Beliefs about capabilities	- Successful strategies and adoption in initially targeted services was replicated in others - Within education sessions and team discussions, clinicians who had adopted the innovation to practice were facilitated to share experiences and tips related to advice or counselling, use of developed materials and consumer feedback	√	P *	√	√	- * Difficult to execute parts with H1 cardiology staff as no follow up team discussions were conducted
Global challenges experienced	- Hospital cardiology services had high patient turnover with time-limited clinicians - Clinics, cardiac rehabilitation programs or routine team meetings were ceased or required remote delivery multiple times due to COVID-19 pandemic - In relation to the COVID-19 pandemic there were significantly increased workload pressures, staff burnout and staff shortages, which may have impacted engagement						

Explanations/abbreviations: ERIC, Expert Recommendations for Implementing Change; TDF, Theoretical Domains Framework; Target service 1 = Hospital 1 Diabetes, 2 = Hospital 1 Cardiology, 3 = Hospital 2 Cardiology, 4 = Community Chronic Disease Service; √, yes were engaged; ×, no were not able to be engaged; P, able to be engaged in part; NA, not applicable; H1, Hospital 1; H2, Hospital 2; CDS, chronic disease service; MDP, Mediterranean-style dietary pattern. * Queensland Health, Nutrition Education Materials Online (NEMO), see https://www.health.qld.gov.au/nutrition/patients accessed on 1 December 2022.

**Table 3 healthcare-13-00506-t003:** Characteristics of eligible clinicians who participated in the survey (n = 57).

Variable	n (%)
Age	
20 to 29 years	12 (21.1)
30 to 39 years	16 (28.1)
40 to 49 years	9 (15.8)
50 to 59 years	13 (22.8)
60 to 69 years	3 (5.3)
Not reported	4 (7.0)
Gender	
Female	39 (68.4)
Male	15 (26.3)
Non-binary/third gender	1 (1.8)
Prefer not to say	2 (3.5)
Region of birth	
Australia	39 (68.4)
Outside Australia	16 (28.1)
Asia	7 (12.3)
United Kingdom	4 (7.0)
Oceania	2 (3.5)
Africa	2 (3.5)
Europe	1 (1.8)
Not reported	2 (3.5)
Mediterranean background *	0 (0.0)
Health professional role	
Dietitian	7 (12.3)
Nurse	16 (28.1)
Nurse Practitioner	6 (10.5)
Diabetes Educator	7 (12.3)
Doctor, Consultant/Senior Medical Officer	12 (21.1)
Doctor, Advanced Trainee/Registrar	2 (3.5)
Doctor, Resident/House officer	1 (1.8)
Occupational Therapist	2 (3.5)
Physiotherapist	1 (1.8)
Psychologist	1 (1.8)
Pharmacist	1 (1.8)
Podiatrist	1 (1.8)
Duration in role	
<1 year	7 (12.3)
1 to 2 years	8 (14.0)
>2 to 5 years	6 (10.5)
>5 to 10 years	6 (10.5)
>10 to 15 years	9 (15.8)
>15 to 20 years	7 (12.3)
>20 years	14 (24.6)
Duration working with coronary heart disease and/or type 2 diabetes patients	
<1 year	4 (7.0)
1 to 2 years	5 (8.8)
>2 to 5 years	5 (8.8)
>5 to 10 years	7 (12.3)
>10 to 15 years	10 (17.5)
>15 to 20 years	10 (17.5)
>20 years	16 (28.1)
Relevant target service/s within past year ^	
Hospital 1 Diabetes	13 (22.8)
Hospital 1 Cardiology	23 (40.4)
Hospital 2 Cardiology	13 (22.8)
Community Chronic Disease Service	14 (24.6)
Clinical setting/s within target services ^	
Inpatient unit/s	30 (54.5)
Outpatient clinic/s	30 (54.5)
Heart recovery, cardiac rehabilitation or heart failure service	21 (38.2)
Community clinic/s	12 (21.1)

* Determined as whether participant reported they or their parents were born in a country bordering the Mediterranean sea. ^ Some clinicians worked across multiple relevant target services and clinical settings.

**Table 4 healthcare-13-00506-t004:** Clinician survey responses related to reach and impact of select implementation strategies.

Practice Variable	Total Cohort (n = 57)	Dietitian (n = 7)	Nurse/Diabetes Educator (n = 29)	Doctor (n = 15)	Other Allied Health (n = 6)	*p*-Value *
Education sessions delivered to clinical teams in routine formats						
Attended an education session						
Yes, in-person or via online meeting	29 (50.9)	3 (42.9)	13 (44.8)	12 (80.0)	1 (16.7)	
Yes, watched a recording	8 (14.0)	3 (42.9)	5 (17.2)	0 (0.0)	0 (0.0)	
No, but colleagues who attended passed on information	7 (12.30)	1 (14.3)	4 (13.8)	1 (6.7)	1 (16.7)	
No	13 (22.8)	0 (0.0)	7 (24.1)	2 (13.3)	4 (66.7)	
If had not attended (n = 20), expressed would be interested in session	16 (80.0)	1 (100.0)	10 (90.0)	1 (33.3)	4 (80.0)	
Expressed having learnt from the session (n = 36) ^						
Yes	36 (100.0)	6 (100.0)	17 (100.0)	12 (100.0)	1 (100.0)	
No or cannot recall	0 (0.0)	0 (0.0)	0 (0.0)	0 (0.0)	0 (0.0)	
No, it only confirmed my existing knowledge	0 (0.0)	0 (0.0)	0 (0.0)	0 (0.0)	0 (0.0)	
Improved knowledge of current dietary evidence/guidelines for heart disease or diabetes	27 (75.0)	5 (83.3)	11 (64.7)	10 (83.3)	1 (100.0)	0.581
Helped understand gaps between dietary evidence and routine care for heart disease or diabetes	20 (55.6)	3 (50.0)	10 (58.8)	6 (50.0)	1 (100.0)	0.777
Improved knowledge of the food principles of a Mediterranean-style or heart healthy dietary pattern	27 (75.0)	5 (83.3)	12 (70.6)	9 (75.0)	1 (100.0)	0.866
Directed to patient education materials	23 (63.9)	3 (50.0)	10 (58.8)	9 (75.0)	1 (100.0)	0.594
Improved confidence to discuss diet with relevant patients	19 (52.8)	3 (50.0)	9 (52.9)	6 (50.0)	1 (100.0)	0.813
Useful for application to own dietary habits	19 (52.8)	3 (50.0)	9 (52.9)	6 (50.0)	1 (100.0)	0.813
Education session changed practice	31 (86.1)	6 (100.0)	13 (76.5)	11 (91.7)	1 (100.0)	0.430
Access of electronic resources provided to clinical teams † (n = 55)						
Yes, at least 1 or more	37 (67.3)	6 (85.7)	18 (66.7)	10 (66.7)	3 (50.0)	0.593
No, do not recall or do not know how to access	11 (20.0)	0 (0.0)	7 (25.9)	2 (13.3)	2 (33.3)	
No, aware exist but have not accessed	7 (12.7)	1 (14.3)	2 (7.4)	3 (20.0)	1 (16.7)	
Copy of the slides presented in education session	23 (41.8)	4 (57.1)	12 (44.4)	6 (40.0)	1 (16.7)	0.506
Copies of patient education materials	28 (50.9)	6 (85.7)	12 (44.4)	9 (60.0)	1 (16.7)	0.067
Publication/s of a particular study	9 (16.4)	1 (14.3)	5 (18.5)	1 (6.7)	2 (33.3)	0.492
Guidelines or evidence summaries	15 (27.3)	4 (57.1)	8 (29.6)	3 (20.0)	0 (0.0)	0.118
Orientation, handover or caseload material related to diet for clinicians who commenced their role in a target service within the past year (n = 22)						
Yes, related to Mediterranean-style or heart-healthy dietary pattern	8 (36.4)	4 (66.7)	3 (33.3)	0 (0.0)	1 (25.0)	0.335
Yes, but other dietary approach	2 (9.1)	1 (16.7)	1 (11.1)	0 (0.0)	0 (0.0)	
Nil related to diet	12 (54.5)	1 (16.7)	5 (55.6)	3 (100.0)	3 (75.0)	

Data are n (%). * Statistical test is Chi-squared test with significance at *p* < 0.05. Data missing for ^ 1 participant (diabetes educator) and † 2 participants (1 nurse, 1 diabetes educator) with incomplete surveys.

**Table 5 healthcare-13-00506-t005:** Clinician survey responses related to the acceptability, feasibility, adoption and sustainability of using a Mediterranean-style or heart-healthy dietary pattern approach * in practice.

Question and Response Options	Total Cohort (n = 57)	Dietitian (n = 7)	Nurse/Diabetes Educator (n = 27)	Doctor (n = 15)	Other Allied Health (n = 6)	*p*-Value **
Questions related to acceptability of approach						
Recommending this would align with scientific evidence (n = 55) †						
Agree/strongly agree	55 (100.0)	7 (100.0)	27 (100.0)	15 (100.0)	6 (100.0)	
Disagree/strongly disagree	0 (0.0)	0 (0.0)	0 (0.0)	0 (0.0)	0 (0.0)	
This approach is too complex to raise or recommend (n = 55) †						NR
Most of the time or always	4 (7.2)	0 (0.0)	2 (7.4)	1 (6.7)	2 (33.3)	
Sometimes	20 (36.4)	1 (14.3)	13 (48.1)	5 (33.3)	2 (33.3)	
Rarely or never	31 (56.4)	6 (85.7)	12 (44.4)	9 (60.0)	2 (33.3)	
This dietary pattern would be appropriate to recommend (n = 55) †						
Agree/strongly agree	52 (94.5)	7 (100.0)	25 (92.6)	15 (100.0)	5 (83.3)	
Disagree/strongly disagree	3 (5.5)	0 (0.0)	2 (7.4)	0 (0.0)	1 (16.7)	
Questions related to feasibility of approach						
Enough time to raise/recommend this in inpatient setting (n = 32) ^						0.402
Most of the time or always	14 (43.8)	4 (100.0)	8 (53.3)	1 (10.0)	1 (33.3)	
Sometimes	9 (28.1)	0 (0.0)	3 (20.0)	5 (50.0)	1 (33.3)	
Rarely or never	9 (28.1)	0 (0.0)	4 (26.7)	4 (40.0)	1 (33.3)	
Enough time to raise/recommend this in outpatient or community setting (n = 39) ^†						
Most of the time or always	23 (59.0)	6 (85.8)	9 (75.0)	7 (46.7)	1 (20.0)	
Sometimes	12 (30.8)	1 (14.3)	3 (25.0)	6 (40.0)	2 (40.0)	
Rarely or never	4 (10.3)	0 (0.0)	0 (0.0)	2 (13.3)	2 (40.0)	
Enough time to raise/recommend this in cardiac rehabilitation (n = 17) ^						0.096
Most of the time or always	10 (58.8)	1 (50.0)	8 (80.0)	1 (100.0)	0 (0.0)	
Sometimes	6 (35.3)	1 (50.0)	2 (20.0)	0 (0.0)	3 (75.0)	
Rarely or never	1 (5.9)	0 (0.0)	0 (0.0)	0 (0.0)	1 (25.0)	
Patients would be able to improve their eating habits to better align with this dietary pattern (n = 55) †						0.097
Most of the time or always	32 (58.2)	5 (61.4)	18 (66.6)	6 (40.0)	3 (50.0)	
Sometimes	21 (38.2)	2 (28.6)	9 (33.3)	7 (46.7)	3 (50.0)	
Rarely or never	2 (3.6)	0 (0.0)	0 (0.0)	2 (13.3)	0 (0.0)	
Questions related to adoption of approach						
Advice would align to these dietary pattern principles (n = 55) †						0.266
Most of the time or always	47 (85.4)	7 (100.0)	23 (85.2)	13 (86.7)	4 (66.7)	
Sometimes	7 (12.7)	0 (0.0)	4 (14.8)	2 (13.3)	1 (16.7)	
Rarely or never	1 (1.8)	0 (0.0)	0 (0.0)	0 (0.0)	1 (16.7)	
Diet-related education materials or tools provided align to these dietary pattern principles (n = 53) ^†						
Most of the time or always	44 (80.0)	7 (100.0)	22 (81.5)	12 (80.0)	3 (75.0)	
Sometimes	8 (14.5)	0 (0.0)	5 (18.5)	3 (20.0)	0 (0.0)	
Rarely or never	0 (0.0)	0 (0.0)	0 (0.0)	0 (0.0)	0 (0.0)	
I don’t know if it would	1 (1.8)	0 (0.0)	0 (0.0)	0 (0.0)	1 (25.0)	
Focus on foods or meals more than nutrients or calories in diet-related assessment, advice or information (n = 54) ^						0.140
Most of the time or always	38 (70.4)	6 (85.7)	21 (72.4)	10 (66.7)	1 (33.3)	
Sometimes	11 (20.4)	1 (14.3)	5 (17.2)	4 (26.7)	1 (33.3)	
Rarely or never	5 (9.3)	0 (0.00	3 (10.3)	1 (6.7)	1 (33.3)	
Focus on what to include more than what to restrict or cut out in diet-related assessment, advice or information (n = 51) ^#						0.519
Most of the time or always	29 (56.9)	6 (85.7)	14 (51.9)	7 (46.7)	2 (66.7)	
Sometimes	16 (31.4)	1 (14.3)	9 (34.6)	6 (40.0)	0 (0.0)	
Rarely or never	6 (11.8)	0 (0.00	3 (14.3)	2 (13.3)	1 (33.3)	
Used at least one or more of the recommended patient education materials (n = 55) †	45 (81.8)	7 (100.0)	23 (85.2)	12 (80.0)	3 (50.0)	0.630
Types of recommended patient education materials						
Related to a Mediterranean-style diet (factsheet or website)	32 (58.2)	7 (100.0)	15 (55.6)	9 (60.0)	1 (16.7)	
Related to a heart healthy dietary pattern (factsheet, National Heart Foundation resources and/or videos)	31 (56.4)	6 (85.7)	16 (59.3)	6 (40.0)	3 (50.0)	
Specific factsheet displayed in clinic rooms						
2-page health service Mediterranean-style diet	29 (52.7)	7 (100.0)	13 (48.1)	9 (60.0)	0 (0.0)	
Heart Foundation heart healthy eating principles Pictorial	18 (32.7)	2 (28.6)	10 (37.0)	4 (26.7)	2 (33.30)	
Healthy convenient meal preparation/pre-prepared meals	18 (32.7)	7 (100.0)	8 (29.6)	3 (20.0)	0 (0.0)	
Perceived sustainability requirements (where practice adopted, what would be important to maintain this) (n = 53) ‡						
Access to hardcopy patient education materials	45 (84.9)	6 (85.7)	21 (80.8)	14 (93.3)	4 (80.0)	0.735
Access to electronic patient education materials	32 (60.4)	5 (71.4)	18 (69.2)	7 (46.7)	2 (40.0)	0.354
Refreshers or updates on evidence	28 (52.8)	6 (85.7)	15 (57.7)	3 (20.0)	4 (80.0)	0.010 **
Refreshers or updates on practical tips/tools	31 (58.5)	6 (85.7)	15 (57.7)	6 (40.0)	4 (80.0)	0.157
Access to or relationship with dietitian	33 (62.3)	3 (42.9)	18 (69.2)	11 (73.3)	1 (20.0)	0.100

Data are n (%). * A summary of core Mediterranean-style or heart healthy dietary pattern principles was provided prior to these questions to inform responses, ** Statistical test is Chi-squared test with significance at *p* < 0.05. ^ Questions only asked of participants who had identified relevant diet-related or clinical roles. # Error in response options for 3 participants not included, and data missing for † 2 participants (1 nurse, 1 diabetes educator) with incomplete surveys; ‡ 3rd participant (nurse) with incomplete survey and 1 allied health professional did not identify as having practice aligned with the approach. NR; not reported as 100% clinicians with same response.

**Table 6 healthcare-13-00506-t006:** Characteristics of eligible patients who participated in the survey (n = 55).

Variable	n (%)
Age	
20 to 29 years	1 (1.8)
30 to 39 years	2 (3.6)
40 to 49 years	6 (10.9)
50 to 59 years	15 (27.3)
60 to 69 years	21 (38.2)
70 to 79 years	9 (16.4)
80 to 89 years	1 (1.8)
Gender	
Male	39 (70.9)
Female	16 (29.1)
Non-binary/third gender	0 (0.0)
Prefer not to say	0 (0.0)
Region of birth	
Australia	35 (63.6)
Outside Australia	20 (36.4)
Asia	2 (3.6)
United Kingdom	3 (5.5)
Oceania	10 (18.2)
Africa	1 (1.8)
Europe	4 (7.3)
Mediterranean background *	6 (10.9)
English second language	11 (20.0)
Relevant diagnosed health condition/s	
Coronary heart disease	41 (74.5)
Type 2 diabetes	28 (50.9)
Both conditions	14 (25.5)
Relevant target service recently accessed ^	
Hospital 1 diabetes	13 (23.6)
Hospital 1 cardiology	31 (56.4)
Hospital 2 cardiology	14 (25.0)
Community Chronic Disease Service	19 (34.5)
Clinical setting/s accessed within target sevices ^	
Inpatient unit/s	28 (50.9)
Outpatient clinic/s	33 (60.0)
Cardiac rehabilitation	22 (40.0)
Community clinic/s	12 (21.8)
Received care from dietitian of target service/s	
No, was not offered	21 (38.2)
No, was offered but declined	2 (3.6)
Yes	32 (58.2)
Inpatient setting only	3 (9.4)
Outpatient clinic only	15 (46.9)
Cardiac rehabilitation only	9 (28.1)
All 3 of the above	2 (6.3)
Inpatient setting and outpatient clinic	1 (3.1)
Inpatient setting and cardiac rehabilitation	2 (6.3)
Whether respondent would have liked to see dietitian (n = 21)	
Yes	11 (47.8)
Unsure how dietitian could help	4 (17.4)
No	6 (26.1)
Received diet-related care from other health professional/s of target service/s	
No	22 (40.0)
Yes	33 (60.0)
Doctor	18 (32.7)
Nurse and/or diabetes educator	26 (47.3)
Allied health professional #	10 (18.2)

* Determined as whether reported by the participant that they or their parents were born in a country bordering the Mediterranean sea. ^ Some patients had recently accessed care from multiple relevant target services and clinical settings. # These included a physiotherapist (n = 9), pharmacist (n = 4), psychologist (n = 2) or social worker (n = 2).

**Table 7 healthcare-13-00506-t007:** Patient survey responses related to the recall of diet-related care in relevant services.

Nature of Diet-Related Care	By Dietitian/s (n = 32)	By Other Health Professional/s (n = 33)
Topics discussed or were taught about that align to principles of a Mediterranean-style or heart-healthy dietary pattern		
A Mediterranean-style diet or way of eating	19 (59.4)	6 (18.2)
Foods to eat for a healthy heart	22 (68.8)	11 (33.3)
Including fruits and vegetables	28 (87.5)	10 (30.3)
Choosing whole grain or brown breads and cereals	26 (81.3)	9 (27.3)
Including legumes	23 (71.9)	7 (21.2)
Including fish/seafood	24 (75.0)	7 (21.2)
Including nuts/seeds	23 (71.9)	6 (18.2)
Including yoghurt, milk or cheese	21 (65.6)	7 (21.2)
Using extra virgin olive oil	20 (62.5)	6 (18.2)
Limiting red meat	21 (65.6)	7 (21.2)
Limiting processed foods or sweets	26 (81.3)	7 (21.2)
Cooking meals or preparing food at home	14 (43.8)	4 (21.1)
Using herbs or spices rather than salt	18 (56.3)	7 (21.2)
Choosing water as your main drink	24 (75.0)	12 (36.4)
At least one or more of the above topics	32 (100.0)	22 (66.7)
Other topics discussed or were educated about		
Limiting intake of carbohydrate	20 (62.5)	7 (21.2)
Limiting intake of fat or saturated fat	23 (71.9)	5 (15.2)
Limiting intake of salt or sodium	22 (68.8)	9 (27.3)
Weight loss or restricting calories	15 (46.9)	9 (27.3)
Managing fluid intake	17 (53.1)	9 (27.3)
Topics of information materials given, recommended or directed that align with Mediterranean-style or heart-healthy dietary pattern		
Provided at least 1 or more of the listed information material	(n = 32)	(n = 33)
Yes	28 (87.5)	23 (69.7)
Provided material but cannot recall details	1 (3.1)	3 (9.1)
No	3 (9.4)	7 (21.2)
Related to a Mediterranean-style diet (factsheet, website and/or book)	20 (62.5)	9 (27.3)
Related to a heart healthy eating pattern (factsheet, Heart Foundation resources and/or videos)	24 (75.0)	18 (54.5)
Related to heart healthy food plate portions	19 (59.4)	9 (27.3)
Related to heart healthy foods for snacks	12 (37.5)	7 (21.2)
Related to healthy convenient meal preparation or pre-prepared meals	9 (28.1)	6 (18.2)
Read/watched relevant information materials provided	(n = 28)	(n = 26)
Yes, all of it	16 (50.0)	11 (42.3)
Yes, some of it	11 (34.4)	12 (46.2)
No	1 (3.1)	3 (11.5)
	For relevant participants	
Exposure to information on Mediterranean-style or heart-healthy dietary pattern in clinic settings *		
On information boards on the ward (n = 28)	11 (39.3)	
TV slides, poster or factsheet in outpatient waiting area (n = 33)	15 (45.5)	
Poster or factsheet in cardiac rehabilitation (n = 22)	14 (63.6)	
Influence of diet-related care (n = 44) ^		
Dietary advice or information taught something new		
Yes, all was new	8 (18.2)	
Yes, some was new	29 (65.9)	
No	7 (15.9)	
Dietary changes as a result of care		
Yes, already made changes	42 (95.5)	
Yes, plan to make changes	2 (4.5)	
No	0 (0.0)	

Data are n (%). * Questions asked only of patients who received care in these settings. ^ In total cohort of participants who received diet-related care from at least one dietitian or other health professional; data missing for 1 participant with incomplete survey.

## Data Availability

The datasets generated and/or analysed during the current study are not publicly available to ensure the confidentiality of the participants, but they are available from the corresponding author on reasonable request and with permissions from the governing ethics committee.

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
