# Peer review of "Translating Evidence for a Mediterranean-Style Dietary Pattern into Routine Care for Coronary Heart Disease and Type 2 Diabetes: Implementation and Evaluation in a Targeted Public Health Service in Australia [Author-notes fn1-healthcare-13-00506]"

_healthcare, 2025, doi:10.3390/healthcare13050506_

Round 1

Reviewer 1 Report (Previous Reviewer 2)

Comments and Suggestions for Authors

The article under review addresses an important and clinically relevant issue by evaluating the implementation of a Mediterranean-style dietary pattern into routine care for coronary heart disease and type 2 diabetes within a targeted public health service. The strategies based on the Knowledge-to-Action cycle and involving multi-disciplinary clinicians provide valuable insights into the practical application of dietary guidelines.

The manuscript has been improved since the first version. However, an additional recommendation at the end of the Discussion section is to include a dedicated section on the novelty of the study and future research directions. This section should specifically address potential barriers to adherence from both patient and clinician perspectives, as these aspects are important for the effectiveness of the intervention. 

Author Response

Reviewer 2 Report (New Reviewer)

Comments and Suggestions for Authors

The manuscript reports the results of an implementation study directed at clinicians to improve MDP's application in treating patients with T2D and CHD. The overall impression of the manuscript was that it was well-written and informative, offering a valuable contribution to the literature. The current manuscript covered the implementation and evaluation parts of the study. While this was clearly explained in the methods, it was somewhat confusing when methods were mixed with the context of the study and the previous results. Authors may wish to ensure that the context of the study and methods are separated.

Further, while this point was touched on in the discussion, the introduction would have benefited from better incorporating previous research examining challenges in implementing dietary advice in clinical practice (e.g., a study from Ireland). This would have also improved the readers' comprehension of the research context (not only the previous study examining barriers and facilitators, but what challenges previous studies have encountered in implementing MDP in clinical practice).

While research aims were clearly articulated, the hypothesis would benefit from further work, as in its current form, this appears instead as a list of aims rather than clear expectations of what would be expected to happen, e.g. increased self-reported knowledge (e.g. clinicians). Further, as patients were also asked as a part of the implementation process, it would be good to add clear rationale, aims (and if relevant, hypotheses) for the patient involvement.

In the results section, a small point regarding Tables 3 and 6. Some of the categories reported are very small, i.e. under 5 participants. This considerably increases the risk (even when still small) that a participant could be identified from the data. Could the authors please consider combining categories with small participant numbers?

The discussion repeated results on occasion, e.g. line 441 “... and 95%...”. Could the authors please check this? Further, regarding the hypotheses, this part of the discussion would benefit from further refinement in line with the previous comments. In addition, considering the length of the discussion, some of the discussions appeared only peripherally relevant to the research aims, e.g., telehealth on line 569 or the discussion from line 570. While these are interesting and valid points, the authors may wish to consider making the discussion more concise.   

Author Response

This manuscript is a resubmission of an earlier submission. The following is a list of the peer review reports and author responses from that submission.

Round 1

Reviewer 1 Report

Comments and Suggestions for Authors

Overall comments: The study aims to summarize the implementation strategies of the Mediterranean-style dietary pattern (MDP) as clinical guidelines for the routine care of coronary heart disease (CHD) and Type 2 Diabetes (T2D). It is the first study that applied implementation science methods to translate scientific evidence of MDP into routine care for people with CHD or T2D. Feedback collected from Multi-disciplinary clinicians in cardiology and diabetes services treating patients with CKD and T2D suggests that MDP can improve health outcomes for patients with CHD and T2D. Though experimental approaches are generally sound and the findings are significant, there are several areas that require attention and clarification.

Major concerns:

1.     Mediterranean-style dietary pattern (MDP) is the most studied dietary pattern worldwide and has broad evidence from randomized controlled trials. Implementation and evaluation of MDP is of clinically significant and obtained essential priority since there is no previous article reported. Thus, current work is with great innovation and significance. However, a Pre-post Implementation Study design cause great concerns. Since the MDP is widely accepted, the current work should focus on higher evidence such as randomized or matched study, so that can make the conclusion more convincible. Although, there is session about comparison with other professionals (table 7), the participants choose and set of blind and random should be detailed in method.

2.     Considering that there were 57 participates from Australian metropolitan public health service sector (7 dietitians, 29 nurses or diabetes educators, 15 doctors and 9 other allied health professionals) were involved in this study, I would like to suggest revise the title to “Translating evidence for the Mediterranean-style dietary pattern into routine care for coronary heart disease and type 2 diabetes:  implementation and evaluation in targeted public health services in Australian”.

3.     What are the characteristics of clinicians or CHD/T2D patients who are more receptive to the implementation strategies of MDP guidelines? For example, factors such as age, gender, or region of birth. Providing more detailed information in this regard would be helpful. Actually, in table 3, we recommended to show the difference between different clinicians. For example, if there is any difference of age or education background between nurse and doctors. And what is the difference between senior and junior clinicians. These are main factors that can influence the implementation.

4.     This study did not demonstrate the benefits of the MDP for CHD or T2D patients. Which clinical indicators showed improvement following the adoption of the MDP guideline implementation strategies? It is recommended to present the detail diet habits and the number of different foods and nutrients. And the evaluation method in table 7 caused great overestimate of the patients if the answer is yes or no but not with dose threshold.

5.     This study included 14 patients with both CHD and T2D (In table 6). However, the implementation of MDP may differ for patients with CHD and T2D compared to those with CHD or T2D alone. It would be valuable if the authors could expand the study to include patients who only suffer from CHD or T2D, and compare the data with those who suffer from both CHD and T2D. Furthermore, all the patients are CHD or T2D patients, it is necessary for authors to explain how did current study decide the effect comes from current implementation other than previous diet education.

Minor concerns

1.     Table 2 is terrible. The column of service engage activity is malposed, which makes the result hard to understand. Symbols like “P” should be noted in footnote.

2.     Majority of references are out of date. It is recommended to refer to last 5 years articles.

Reviewer 2 Report

Comments and Suggestions for Authors

This is an interesting study that addresses a gap in translating evidence-based dietary guidelines into routine clinical care, which is highly relevant for managing coronary heart disease and type 2 diabetes. However, I have a few comments on the manuscript:

1. The limited sample size of clinicians (n=57) and patients (n=55) necessitates classifying this study as preliminary. The manuscript's title and abstract should explicitly indicate this to manage readers’ expectations and appropriately contextualize the findings.

2. The abstract lacks specific details on the content and structure of the implementation strategies. For instance, what specific topics were covered in the education sessions? How were the educational materials tailored to the local context? A brief mention of these specifics would improve the clarity of the intervention.

3. The Materials and Methods section requires restructuring for clarity and comprehensiveness. Study Design: Clearly describe the implementation and evaluation phases. Participants: Provide detailed information on clinician and patient recruitment, inclusion/exclusion criteria, and justification of the sample size. Additionally, a flow chart visualizing the selection and inclusion of participants is recommended to enhance transparency.

4. The study are the primary evaluation tool; however, their design and validity are not addressed. Were these surveys validated tools, or were they developed specifically for this study? Including such details would strengthen the methodological section.

5. The results section focus on the success of the implementation but lack a comparison with baseline data (e.g., pre-implementation knowledge or practices). Including such comparisons would better illustrate the extent of change. Furthermore, Tables 4, 5, and 7 lack detailed statistical analyses. The manuscript should specify the statistical tests used for intergroup comparisons.

6. At the end of the Discussion section, it is recommended to include a separate subsection titled "Strengths and Limitations." Additionally, at the conclusion of the manuscript, it is advised to further emphasize the implications and novelty of this study.